# TacDexGrasp: Compliant and Robust Dexterous Grasping with QP and Tactile Feedback

**Yubin Ke**[1,2*], **Jiayi Chen**[1,2*], **Hang Lv**[1,2*], **Xiao Zhou**[1,2], **He Wang**[1,2†]
[1]Peking University. [2]Galbot.
*Equal contribution. †Corresponding author: hewang@pku.edu.cn.

**Abstract:** Multi-fingered hands offer great potential for compliant and robust grasping of unknown objects, yet their high-dimensional force control presents a significant challenge. This work addresses two key problems: (1) distributing contact forces to counteract an object's weight, and (2) preventing rotational slip caused by gravitational torque when a grasp is distant from the object's center of mass. We address these challenges via tactile feedback and a quadratic programming (QP)-based controller, without explicit torque modeling or slip detection. Our key insights are (1) rotational slip will induce translational slip for a multi-fingered grasp, and (2) the ratio of tangential to normal force at each contact is an effective early stability indicator. By actively constraining this ratio for each finger below the estimated friction coefficient, our controller maintains grasp stability against both translational and rotational slip. Real-world experiments on 12 diverse objects demonstrate the robustness and compliance of our approach.

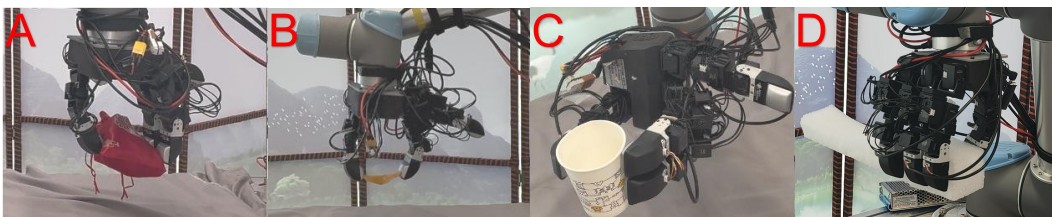

Figure 1: **TacDexGrasp.** (A-C) Our system enables stable and safe grasping for multi-fingered hands on objects with diverse, unknown mass distributions, friction coefficients, and deformation materials. (D) Our QP-based controller can also compensate for gravitational torque to prevent the rotational slip without explicit torque modeling.

## 1 Introduction

Applying appropriate forces is essential for both stability and safety in robotic grasping. Excessive force may damage delicate objects, while insufficient force can lead to slippage or grasp failure. Prior work on compliant grasping [1, 2, 3, 4, 5, 6, 7, 8] has largely focused on parallel grippers or highly underactuated multi-fingered hands. These designs simplify grasp synthesis and control by reducing the number of degrees of freedom (DoF), but at the cost of dexterity and adaptability in complex tasks. For example, parallel grippers are particularly limited in preventing rotational slip caused by gravitational torque when a grasp is offset from an object's center of mass.

In contrast, fully actuated dexterous hands—with their human-like morphology—hold greater potential for versatile and robust grasping. However, their increased DoF and more complex contact interactions introduce a new challenge in how to distribute forces across multiple fingers to support an object's weight. Moreover, it is also underexplored how to prevent the rotational slip by a multi-fingered hand.

Submitted to the 8th Conference on Robot Learning (CoRL 2024). Do not distribute.

In this work, we tackle these challenges using tactile feedback and a quadratic programming (QP)-based force controller, without relying on torque modeling or slip detection. Our first key insight is that rotational slip of the object inevitably induces translational slip at some contacts in a multi-fingered grasp. This means that explicitly modeling gravitational torque is unnecessary; preventing translational slip at each finger is sufficient for stability.

Our second observation is that the ratio of tangential to normal force at a contact provides an early and reliable stability indicator. By constraining this ratio below the estimated friction coefficient, slip can be avoided before it occurs. This constraint can be naturally integrated into a QP formulation, which we solve efficiently in around 2 ms to compute the optimal force distribution across contacts. The resulting target forces are then tracked using a PID controller on the robot hand.

We validate our method in real-world experiments on 12 objects with diverse, unknown mass distributions, friction coefficients, and deformation materials. Our approach achieves a 83% grasp success rate without damaging delicate objects, outperforming prior baselines. Furthermore, our controller demonstrates robustness by adapting to unexpected external disturbances during grasping.

## 2 Method

### 2.1 Solving Contact Force by Quadratic Programming

In this section, we introduce our QP formulation to compute the target force for each contact to balance the object's weight, while constraining the ratio of tangential to normal force to prevent slip.

We consider an object $O$ grasped at $m$ contact points. For each contact $i$, $\mathbf{n}_i \in \mathbb{R}^3$ is the inward surface normal, while $\mathbf{d}_i, \mathbf{c}_i \in \mathbb{R}^3$ are orthogonal tangents with $\mathbf{n}_i = \mathbf{d}_i \times \mathbf{c}_i$, all expressed in the world frame. The Coulomb friction cone $\mathcal{F}_i$ and local-to-world transformation $\mathbf{J}_i$ are defined as

$$\mathcal{F}_i = \left\{ \mathbf{x}_i \in \mathbb{R}^3 \mid 0 \le x_{i,1} \le \gamma, x_{i,2}^2 + x_{i,3}^2 \le \mu^2 x_{i,1}^2 \right\}, \quad \mathbf{J}_i^T = \begin{bmatrix} \mathbf{n}_i & \mathbf{d}_i & \mathbf{c}_i \end{bmatrix} \in \mathbb{R}^{3 \times 3} \tag{1}$$

Here, $\gamma$ is the force upper bound and $\mu$ is the friction coefficient.

To balance the object's weight, the target contact forces $(\mathbf{f}_1^t, ..., \mathbf{f}_m^t)$ at timestep $t$ are obtained from

$$(\mathbf{f}_1^t, ..., \mathbf{f}_m^t) = \underset{(\mathbf{x}_1,...,\mathbf{x}_m)}{\arg\min} \quad \| \sum_{i=1}^{m} \mathbf{J}_{o,i}^T \mathbf{x}_i - \mathbf{g} \|^2 + \beta_1 \sum_{i=1}^{m} \| \mathbf{x}_i^t - \mathbf{f}_i^{t-1} \|^2 + \beta_2 \sum_{i=1}^{m} \| \mathbf{x}_i^t \|^2 \tag{2}$$

$$\text{s.t.} \quad \mathbf{x}_i \in \mathcal{F}_i, \quad i \in \{1, ..., m\} \tag{3}$$

where $\mathbf{g} \in \mathbb{R}^3$ is the object gravity, $\sum_{i=1}^{m} \| \mathbf{x}_i^t - \mathbf{f}_i^{t-1} \|^2$ is the temporal smooth term, and $\sum_{i=1}^{m} \| \mathbf{x}_i^t \|^2$ is the penalty term used to avoid large force. $\beta_1$ and $\beta_2$ are hyperparameters. Similar to [9, 10], approximating the cone constraints to a pyramid yields a linearly-constrained QP, which can be efficiently solved by Clarabel [11] via `qpsolvers` [12].

However, the above QP formulation is not enough to prevent the slippage. Because the tangential contact force acts like a "passive effect" of the normal contact force, which is not directly controllable. We can only use a PID controller to track the target normal force. As a result, the real tangential force can be much larger than the one solved by QP and may violate the cone constraint. To address this issue, we leverage the tactile feedback and update the lower bound of normal force as

$$\mathcal{F}_i = \left\{ \mathbf{x}_i \in \mathbb{R}^3 \mid \frac{\sqrt{(f_{i,2}^r)^2 + (f_{i,3}^r)^2}}{\mu} \le x_{i,1} \le \gamma, x_{i,2}^2 + x_{i,3}^2 \le \mu^2 x_{i,1}^2 \right\} \tag{4}$$

where $f_{i,2}^r$ and $f_{i,3}^r$ are the tangential components of the real contact force obtained from the tactile sensor. Note that we still need to solve $x_{i,2}$ and $x_{i,3}$ to ensure that there is a valid friction combination to balance the object's weight.

### 2.2 Tactile-based Compliant and Robust Dexterous Grasping

In this section, we introduce our system design for compliant and robust dexterous grasping with tactile feedback. The pseudocode is shown in Algorithm 1. Our system consists of three stages:

**Algorithm 1** Tactile-based Compliant and Robust Dexterous Grasping

---

**Data:** $PC$
**Param:** model, $n_1$, $n_2$, $\Delta q_{transport}$, $\mu_{init}$, $G_{init}$
**Init:** $\mu^0 \leftarrow \mu_{init}$, $G^0 \leftarrow G_{init}$, $t \leftarrow 1$
1: $q_{pregrasp}$, $\Delta q_{grasp} \leftarrow$ model($PC$)  ▷ 1. Prediction stage
2: **Move**($q_{pregrasp}$)  ▷ 2. Grasping stage
3: **for** $i = 1$ to $n_1$ **do**  ▷ Squeeze non-contacting fingers
4:   $mask_{finger} \leftarrow$ **GetNonContactFinger**()
5:   **MoveDelta**($\Delta q_{grasp} \cdot mask_{finger}$)
6: **end for**
7: **while** True **do**  ▷ 3. Transport stage
8:   **MoveDelta**($\Delta q_{transport}$)  ▷ Non-blocking arm control
9:   **for** $i = 1$ to $n_2$ **do**  ▷ Adaptive hand control
10:    $n^t$, $f^t_{current} \leftarrow$ **ReadTactile**()
11:    $\mu^t$, $G^t \leftarrow$ **Update**($\mu^{t-1}$, $G^{t-1}$, $f^t_{current}$)
12:    $f^t_{target} \leftarrow$ **QP**($n^t$, $\mu^t$, $G^t$, $f^{t-1}_{target}$)
13:    **ContactForceController**($f^t_{current}$, $f^t_{target}$)
14:    $t \leftarrow t + 1$
15:   **end for**
16: **end while**

---

**Prediction Stage (in line 1):** This stage predicts grasp poses from the partial observation following [9, 10]. Given a single-view segmented object point cloud denoted as $PC$, we use a learned network to predict a pregrasp pose $q_{pregrasp}$ that remains a margin with the object, a delta pose $\Delta q_{grasp}$ that drives the hand to touch the object.

**Grasping Stage (in line 2-6)** This stage controls the hand to establish contact with the object. The hand is first moved to the predicted pre-grasp pose $q_{pregrasp}$, and then squeezed to touch the object over $n_1$ steps. At each step, only those fingers that are not in contact with the object are squeezed, whose mask is obtained from the tactile signals. This strategy enables a soft and progressive approach, minimizing the risk of applying excessive initial force to the object.

**Transport Stage (in line 7-16)** In this stage, the hand adjusts the grasping force in a closed-loop manner while moving the object to follow a predefined trajectory by the robot arm. In most of our experiments, the predefined trajectory first gradually lifts the object and then optionally performs irregular movements. Our adaptive force control for each time $t$ is performed as follows:

- Read tactile data to extract contact normals $n^t$, and actual contact forces $f^t_{current}$.

- Update the estimated gravity and friction coefficient. In each timestep $t$, we first estimate the current value using the tactile data as

$$\hat{\mu}^t = \underset{i}{\text{mean}}\ \frac{f^t_{i,t}}{f^t_{i,n}},\ \hat{G}^t = \frac{g}{g+a}\sum_i f^t_i, \tag{5}$$

  where $f^t_{i,t}$ and $f^t_{i,n}$ denote the tangential and normal components of the contact force at contact point $i$, respectively. $g$ is 9.8 N/kg, and $a$ is the object's acceleration, which we estimate from the acceleration of the robot hand. We use a maximum sliding window filter and an average sliding window filter on $\hat{\mu}^t$ and $\hat{G}^t$ to get the final $\mu^t$ and $G^t$,

- Using the QP formulation introduced in Section 2.1 to compute the target contact forces.

- Using a joint position-based PID controller to track the target forces solved by QP:

$$q_{control} = q_{current} + k_p \tau + k_i \int \tau dt + k_d \dot{\tau} \tag{6}$$

  where $\tau = J^T(q_{current})(f_{target} - f_{current})$ is the error torque.

## 3 Real World Experiments

As shown in Figure 2, 12 different objects with varied deformation materials, friction coefficients, and mass distributions, are used in our experiment. The object mass ranges from 20g to 200g,and the friction coefficients range from 0.4 to 1.2, measured by gradually opening from a parallel grasping and recording the force when the object falls.

We use a 16-DoF Leap hand mounted on a 6-DoF UR5e robotic arm, and an Azure Kinect sensor to capture the RGB and depth images of the scene, while four visual-tactile sensors Tac3D [13] are attached to the fingertips of the Leap hand to provide tactile feedback.

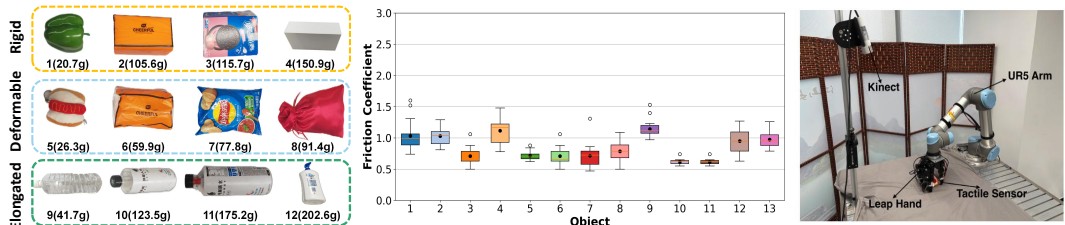

Figure 2: **Real World Setup.** (Left) 12 diverse objects, ordered by their mass. (Middle) Friction coefficients of each object. Note that the mass and friction coefficients are only presented as statistics and not used in our experiments. (Right) Our real robots.

Our approach achieves 83% success rate on these objects, each with 5 trials. As a comparison, naively applying a pre-computed squeeze pose [10] only gives a success rate of approximately 72%, while exerting about 45% more force than our method.

We also show some representative trial on four easily deformable objects in Figure 3, where the target contact forces from the QP formulation are compared against the actual forces measured by the tactile sensors. The QP targets begin with a low initial value and are gradually increased based on tactile feedback, demonstrating our method's ability to achieve stable and safe grasps across objects with varying masses and friction coefficients, while avoiding excessive force that could cause deformation. Furthermore, the actual forces closely follow the QP targets, confirming that our controller effectively regulates the applied force.

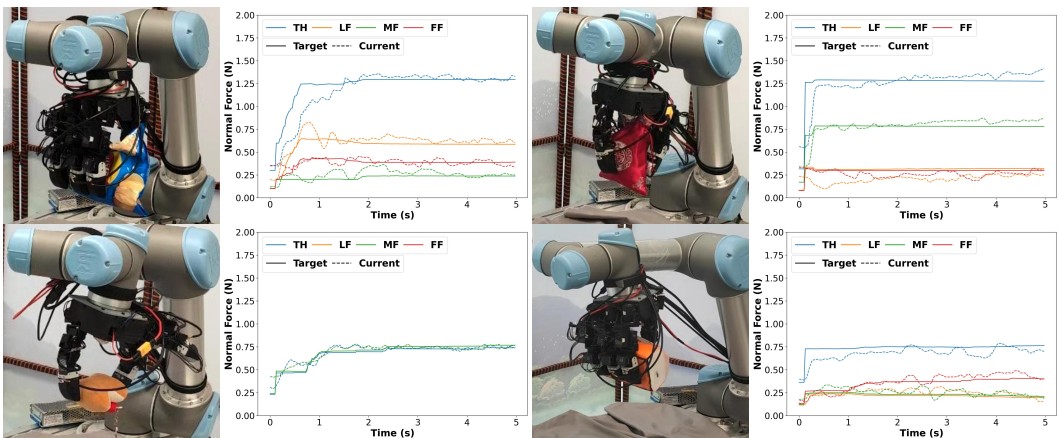

Figure 3: **Real World Experiments.** For diverse objects with different masses, deformation materials, and friction coefficients, our system quickly adapts and performs stable and safe grasps. The real contact force matches well with the target force solved by QP.

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
