# OpenReview forum: "TacDexGrasp: Compliant and Robust Dexterous Grasping with QP and Tactile Feedback"
_robot-learning.org/CoRL/2025/Workshop/Dexterous_Manipulation — CoRL 2025 Workshop Dexterous Manipulation Spotlight_

### Official Review · Reviewer_yoKq · 2025-09-07
**A novel control-based tactile QP formulation for dexterous grasping with robust results, though limited in learning contribution.**

**Rating:** 7
**Confidence:** 3

**Review:**

### Summary:
The paper aims to achieve stable grasping of objects with diverse and unknown mass distributions, friction coefficients, and deformable materials using a multi-fingered dexterous hand. It leverages tactile feedback and quadratic programming (QP) to achieve this goal, without relying on explicit torque modeling or slip detection. The system begins with a learned network that, given a segmented object point cloud, predicts a pre-grasp pose near the object and a delta grasp motion that drives the hand into contact. In the grasping stage, the hand moves to the pre-grasp pose and then progressively closes fingers until contact is detected through tactile signals, ensuring a soft and compliant initial grasp. In the transport stage, the hand executes a predefined trajectory, including lifting and occasional irregular movements, while adaptively updating grasp forces through the proposed QP formulation. The topic is well aligned with the workshop’s goals, as it focuses on tactile feedback and dexterous multi-fingered grasping. The approach primarily addresses the problem from a control perspective rather than a learning-based one.

###  Strengths:
1. A key strength of the paper is its novel formulation that avoids explicit torque modeling and slip detection, which are common in prior work. Instead, the authors directly integrate tactile feedback into their QP formulation.
2. The paper presents an efficient QP formulation that prevents slips by constraining the tangential-to-normal force ratio at each contact to remain below the estimated friction coefficient, ensuring stability against both translational and rotational slip.
3. The method is validated on 13 diverse real-world objects with significant variation in mass and friction coefficients, demonstrating applicability in practical scenarios.

### Weaknesses:
1. The role of learning in the system is minimal, limited to grasp pose prediction, whereas the main contribution lies in a control-based formulation.
2. The explanation of why torque modeling and slip detection are avoided could be expanded. While the authors argue this simplification is sufficient for stability, it remains unclear whether this makes the system more generalizable, computationally lighter, or easier to deploy compared to existing approaches.
3. The explanation of the key insight — that rotational slip inevitably induces translational slip at some contacts, making translational slip prevention sufficient for stability — could be expanded. This is an interesting claim, but it would benefit from a more detailed justification or empirical analysis.

### Questions:
1. Can the authors elaborate further on the points raised in the weaknesses, particularly the rationale for avoiding torque modeling and slip detection, and the justification for the claim that rotational slip inevitably induces translational slip?
2. Is this method applicable to more complex tasks, such as tool use, where a grasped object is employed to manipulate another object?
3. With the recent advancements in learning-based techniques, how might this approach be combined with learning methods to enhance generalization and adaptability?

Overall, I find this paper a strong candidate for the workshop that has sound contributions. While the role of learning is limited, the control perspective is well-motivated, and the experimental results are convincing. Given the workshop’s focus on the intersection of learning and control, it would be valuable to see a discussion of how this control-based formulation might complement learning-based policies.

---

### Official Review · Reviewer_Tvij · 2025-09-10
**Review for Submission Number 20**

**Rating:** 7
**Confidence:** 3

**Review:**

**[Summary]**

This paper presents TacDexGrasp, a system for compliant and robust grasping with multi-fingered hands. It leverages tactile feedback and a quadratic programming (QP)-based controller to stabilize objects with unknown mass distributions, friction coefficients, and material properties.

**[Strengths]**

The method is validated in real-world experiments with 13 diverse objects, achieving high grasp success rates without damaging deformable materials. The integration of tactile feedback with QP optimization provides both compliance and robustness in handling external disturbances.

**[Weaknesses]**

The evaluation is limited to relatively light objects (20–200g), leaving scalability to heavier or more complex shapes uncertain. In addition, the system depends on specialized tactile sensors, which may restrict general applicability in broader settings.

**[Overall Feedback]**

This work is well aligned with the workshop theme, addressing the challenge of robust grasping without explicit torque modeling or slip detection. The contributions are clear and practically relevant, though broader validation would further strengthen the impact.

---

### Decision · Program_Chairs · 2025-09-18

Accept (Spotlight)